# PF127 Hydrogel-Based Delivery of Exosomal CTNNB1 from Mesenchymal Stem Cells Induces Osteogenic Differentiation during the Repair of Alveolar Bone Defects

**DOI:** 10.3390/nano13061083

**Published:** 2023-03-16

**Authors:** Longlong He, Qin Zhou, Hengwei Zhang, Ningbo Zhao, Lifan Liao

**Affiliations:** 1Key Laboratory of Shaanxi Province for Craniofacial Precision Medicine Research, College of Stomatology, Xi’an Jiaotong University, Xi’an 710004, China; 2Department of Implant Dentistry, College of Stomatology, Xi’an Jiaotong University, Xi’an 710004, China

**Keywords:** temperature-responsive hydrogel, bone marrow mesenchymal stem cells, exosomes, CTNNB1, microRNA-146a-5p, IRAK1, TRAF6, alveolar bone defect, osteogenic differentiation

## Abstract

Pluronic F127 (PF127) hydrogel has been highlighted as a promising biomaterial for bone regeneration, but the specific molecular mechanism remains largely unknown. Herein, we addressed this issue in a temperature-responsive PF127 hydrogel loaded with bone marrow mesenchymal stem cells (BMSCs)-derived exosomes (Exos) (PF127 hydrogel@BMSC-Exos) during alveolar bone regeneration. Genes enriched in BMSC-Exos and upregulated during the osteogenic differentiation of BMSCs and their downstream regulators were predicted by bioinformatics analyses. CTNNB1 was predicted to be the key gene of BMSC-Exos in the osteogenic differentiation of BMSCs, during which miR-146a-5p, IRAK1, and TRAF6 might be the downstream factors. Osteogenic differentiation was induced in BMSCs, in which ectopic expression of CTNNB1 was introduced and from which Exos were isolated. The CTNNB1-enriched PF127 hydrogel@BMSC-Exos were constructed and implanted into in vivo rat models of alveolar bone defects. In vitro experiment data showed that PF127 hydrogel@BMSC-Exos efficiently delivered CTNNB1 to BMSCs, which subsequently promoted the osteogenic differentiation of BMSCs, as evidenced by enhanced ALP staining intensity and activity, extracellular matrix mineralization (*p* < 0.05), and upregulated RUNX2 and OCN expression (*p* < 0.05). Functional experiments were conducted to examine the relationships among CTNNB1, microRNA (miR)-146a-5p, and IRAK1 and TRAF6. Mechanistically, CTNNB1 activated miR-146a-5p transcription to downregulate IRAK1 and TRAF6 (*p* < 0.05), which induced the osteogenic differentiation of BMSCs and facilitated alveolar bone regeneration in rats (increased new bone formation and elevated BV/TV ratio and BMD, all with *p* < 0.05). Collectively, CTNNB1-containing PF127 hydrogel@BMSC-Exos promote the osteogenic differentiation of BMSCs by regulating the miR-146a-5p/IRAK1/TRAF6 axis, thus inducing the repair of alveolar bone defects in rats.

## 1. Introduction

The alveolar bone is defined as the component of the maxilla and mandible that provides support for the teeth, with undefined morphological changes and molecular mechanisms [1,2]. The disruption of “coupled” osteoclast–osteoblast actions by chronic periodontal inflammation can ultimately lead to alveolar bone destruction [3]. The regeneration of alveolar bone is pivotal for treating periodontal diseases [4]. Increasing attention has been paid to cell-based bone tissue engineering methods for reconstructing alveolar bone damage, which combines the use of stem cells and biocompatible scaffolds [5].

Hydrogels are known as water-swollen networks, formed from naturally derived or synthetic polymers, which hold a high potential for medical applications and play a crucial role in tissue repair and remodeling [6]. Reference [7] developed the supercritical gel drying of Ch/G mixtures to produce aerogels with improved structural organization and properties in regard to the starting single biopolymers [1,7]. In addition, reference [8] illustrated that the PTMC/PLA/HA and PTMC/HA scaffolds can be prepared utilizing PTMC/PLA/HA and PTMC/HA composite materials, respectively, via the biological 3D printing method and developed as potential biomaterials for bone repatriation and tissue engineering [8]. Nowadays, the thermo-responsive pluronic F127 (PF127) hydrogel has been highlighted to be effective in preserving alveolar bone, which can function as a local drug delivery system for clinical management of periodontitis and related pathologies [9]. It has been suggested that the use of mesenchymal stem cells (MSCs) can favor regeneration in experimental alveolar bone defects [10]. Bone marrow MSCs (BMSCs) are widely applied for bone regeneration because of their self-renewal and differentiating capacities into osteogenic or chondrogenic lineages [11]. Notably, the combination of BMSCs with PF127 was suggested as a promising novel approach to alveolar bone regeneration [12]. Exosomes (Exos) are identified as cell-secreted nanosized small extracellular vesicles (EVs) that can deliver bioactive substances to participate in physiological and pathological processes in the body [13]. A hydrogel loaded with BMSC-derived small EVs resulted in less alveolar bone loss to induce periodontal regeneration [14].

It should be noted that our bioinformatics analysis predicted beta-catenin (CTNNB1) as a pivotal gene both enriched in BMSC-Exos and upregulated in the osteogenic differentiation of BMSCs. As a multitasking and evolutionary conserved molecule, β-catenin (Armadillo in Drosophila) in metazoans plays a pivotal role in multiple developmental and homeostatic processes and functions as an important nuclear effector of canonical Wnt signaling in the nucleus [15]. Interestingly, upregulation of CTNNB1 by diosgenin could enhance bone formation to achieve anti-osteoporotic function [16]. CTNNB1, acting as a transcription factor, was reported to activate microRNA (miR)-146a transcription [17]. miR-146a-5p was identified as a key miR promoting osteogenic differentiation [18]. Human umbilical cord MSC-derived exosomal miR-146a-5p could repress the interleukin-1 receptor associated kinase 1 (IRAK1)/tumor necrosis factor (TNF) receptor associated factor 6 (TRAF6) axis to regulate neuroinflammation [19]. Of note, reference [20] demonstrated that more notable peri-implant bone loss and osteoclastogenesis were found in diabetic mice with glycemic fluctuation, and glycemic fluctuation could result in increases in the expression of IRAK1 and TRAF6 in peri-implant gingival tissues, which suggested that activation of the IRAK1/TRAF6 axis by glycemic fluctuation may contribute to the aggravation of bone loss [20].

Considering the aforementioned evidence, we proposed a hypothesis in this study that PF127 hydrogel loaded with BMSC-Exos (PF127 hydrogel@BMSC-Exos) overexpressing CTNNB1 might affect the repair of alveolar bone defects by regulating the miR-146a-5p-mediated IRAK1/TRAF6 axis (Figure 1).

## 2. Materials and Methods

### 2.1. Ethics Statement

The current study was approved by the Animal Ethics Committee of our institute (protocol number: NO-SH1H-AE-20220127-012). Extensive efforts were made to minimize both the number of animals and their respective suffering in the experiments.

### 2.2. Bioinformatics Analysis

Proteomics dataset PXD020948 related to human BMSC-Exos was obtained from the PRIDE partner database. Microarray GSE9451 (BMSCs group: *n* = 3; BMSCs osteogenesis group: *n* = 3) was obtained from the GEO database, with log*FC* > 1 and *p* < 0.05 used as screening criteria. Significantly upregulated genes were screened using the limma package in R language, and the heatmap was drawn using the pheatmap package in R language. Candidate genes were obtained using the jvenn tool. Protein interaction analysis of the candidate genes was performed in the STRING database and further imported into Cytoscape software (v3.8.2, National Institute of General Medical Sciences, Bethesda Softworks, Rockville, MD, USA) for visualization. The downstream target genes of miR-146a-5p were predicted using the miRDB, TargetScan, and DIANA TOOLS databases.

### 2.3. Culture of Rat BMSCs

Rat BMSCs (R7500, ScienCell, Carlsbad, CA, USA) were cultured with proliferation medium (PM) containing DMEM (Gibco, Grand Island, NY, USA), 10% (*v*/*v*) FBS, 100 U/mL penicillin G, and 100 mg/mL streptomycin (HyClone Company, Logan, UT, USA) in a 5% CO_2_ incubator at 37 °C with 100% relative humidity.

In addition, BMSCs were seeded in a 6-well culture dish at the density of 2 × 10^4^ cells/well. When the cell confluence reached 80%, cells were further cultured with PM supplemented with 5 mM β-glycerol phosphate (G9422, Sigma-Aldrich, St. Louis, MO, USA), 50 μg/mL ascorbic acid (BP461, Sigma-Aldrich, St. Louis, MO, USA), and 10 nM osteogenic medium (OM) containing dexamethasone for osteogenic induction.

### 2.4. Cell Grouping and Plasmid Transfection

Plasmids were purchased from GenePharma (Shanghai, China). BMSCs were seeded in a 6-well plate, and 10 μL Lipofectamine™ 2000 (11068500, Invitrogen, Carlsbad, CA, USA) and 4 μg plasmids were diluted in 150 μL Opti-MEM (11058021, Thermo Fisher Scientific, Rockford, IL, USA) under 70% cell confluence, followed by resting for 20 min and culture with 5% CO_2_ at 37 °C. After 6 h of incubation, the cells were further cultured with complete medium for 48 h. Transfection efficiency was determined by RT-qPCR and Western blot analysis. Cells were transfected with overexpression vector of CTNNB1 (oe-CTNNB1), miR-146a-5p inhibitor, shRNA targeting IRAK1 (sh-IRAK1), and corresponding negative controls (NCs, oe-NC, inhibitor NC, or sh-NC) alone or in combination. The shRNA sequences are shown in Appendix A.

### 2.5. Isolation and Characterization of Exos from BMSCs

Exos were isolated from the supernatant of BMSCs medium by differential centrifugation method [21]. Briefly, after incubation for 48 h in BMSCs medium containing Exos-free FBS, cell supernatants were collected and successively centrifuged at 800× *g* for 5 min to remove dead cells, at 1500× *g* for 15 min to remove cell debris, and at 15,000× *g* for 30 min to remove large EVs. The resulting supernatants were subsequently ultracentrifuged at 150,000× *g* for 2 h, followed by identification of purified Exos.

The morphology of Exos was observed using a transmission electron microscope (TEM) (H-7650, HITACHI, Tokyo, Japan) [22]. The size of Exos was analyzed by nanoparticle tracking analysis (NTA) using a NanoSight LM10 instrument (NanoSight, Wiltshire, UK) [23]. Furthermore, the protein expression of Exos surface markers was assessed by Western blot analysis with such antibodies as rabbit polyclonal antibodies against TSG101 (ab125011, 1:1000, Abcam, Cambridge, UK), CD9 (CBL162, 1:500, Sigma-Aldrich, St. Louis, MO, USA) and Calnexin (ab133615, 1:1000, Abcam, Cambridge, UK) [24].

### 2.6. Construction of Composite Material of Hydrogel Loading with Exos

Pre-cooled PF-127 powder (P2443, Sigma-Aldrich, St. Louis, MO, USA) and 300 μg/mL of aforementioned BMSC-Exos suspension were mixed in a 1.5 mL centrifuge tube. The powder was fully dissolved by vortexing, followed by an ice bath and preservation in a 4 °C refrigerator. The centrifuge tube was checked for precipitation after 1 h. After the precipitation was completely dissolved, the complex of BMSC-Exos and PF-127 hydrogel, namely, PF127 hydrogel@BMSC-Exos, was finally generated. Since PF-127 has a unique temperature sensitivity, and its initial gelation temperature decreases with increasing concentration, we selected the appropriate concentration by determining the time required at both the initial gel temperature and 37 °C based on the 20–32% gradient concentration.

At water bath temperature of 10 °C, the initial gelation temperature of PF127 hydrogel@BMSC-Exos solution at different concentrations (10%, 24%, 28%, and 32%) was tested within the temperature range of 10–40 °C. The specific gelatinization time at 37 °C was determined with the concentration gradient of PF127 hydrogel@BMSC-Exos. The concentration was determined by adjusting another thermostat water bath to 37 °C and observing the specific gelation time of PF127 hydrogel@BMSC-Exos at 37 °C at different concentrations.

### 2.7. Uptake of Exos by BMSCs

Exos were labeled with the PKH67 kit (MINI67-1KT, Sigma-Aldrich, St. Louis, MO, USA), in accordance with the instructions of the manufacturer, and restored at 4 °C for backup. Hydrogel at the above concentration was fully mixed with BMSC-Exos, and the mixture was stored at 4 °C. The BMSCs seeded in 12-well plates were treated with PF127 hydrogel, NC-Exos (BMSC-NC-Exos), PF127 hydrogel@BMSC-NC-Exos), or PF127 hydrogel@BMSC-Exos overexpressing CTNNB1 (PF127 hydrogel@BMSC-CTNNB1) and then cultured with 5% CO_2_ in the incubator for 12 h. After nuclear staining with DAPI (1:1000, Beyotime, Beijing, China), images were captured using a fluorescence microscope.

### 2.8. Alkaline Phosphatase (ALP) Activity Assessment

BMSCs were seeded in a 96-well plate at a density of 1 × 10^5^ cells/well. After 7 d of PM or OM induction, ALP activity was measured at 405 nm using ALP detection kits (A059-3-1, Nanjing Jiancheng Bioengineering Institute, Nanjing, China). The ALP in the cells was also colorized using a color development kit.

### 2.9. Alizarin Red Staining

The BMSCs were transferred to a 24-well plate at a density of 2 × 10^5^ cells/well. At 14 d after PM or OM induction, Alizarin red staining was performed, in accordance with the protocol of the kit (A5533, Sigma-Aldrich, St. Louis, MO, USA). Alizarin red (0.2%, pH = 8.3, Sigma-Aldrich) staining was performed for 10 min at 24–26 °C, and the matrix mineralization level was observed under an inverted microscope. To evaluate the concentration of calcium deposition, Alizarin red dye in BMSCs was extracted with 400 μL 10% (*w*/*v*) sodium chloride solution in 10 mM sodium phosphate solution for 10 min, followed by quantification on a UV visible spectrometer at 562 nm.

### 2.10. Chromatin Immunoprecipitation (ChIP)

Two CTNNB1 shRNA sequences were designed, and the one with optimal knockdown efficiency was selected by RT-qPCR for ChIP. The enrichment of CTNNB1 at the promoter region of miR-146a-5p was detected using the ChIP kit (KT101-02, Saicheng Biotech Co., Ltd., Guangzhou, China). Briefly, under cell confluence of 70–80%, 1% formaldehyde was added to fix the cells at room temperature for 10 min, and the intracellular DNA was crosslinked with protein. Sonication was performed to produce DNA fragments of 300–1000 bp. Immunoprecipitation was performed with the target protein-specific rabbit antibody against CTNNB1 (ab32572, 1:250, Abcam) and with Protein G Dynabeads (Invitrogen). The endogenous DNA–protein complex was precipitated using Protein Agarose/Sepharose, and the supernatant was aspirated after brief centrifugation. After de-crosslinking at 65 °C overnight, the DNA fragments were retrieved through phenol/chloroform purification for reverse transcription-quantitative polymerase chain reaction (RT-qPCR) detection of the miR-146a-5p gene promoter fragment. The primer sequences specific for the promoter region of miR-146a-5p are shown in Appendix A.

### 2.11. Dual-Luciferase Reporter Gene Assay

The binding sites of miR-146a-5p to IRAK1 and TRAF6 were predicted through an online database. The gene fragment of the 3′UTR region of IRAK1 was subjected to clonal amplification, and the PCR product was cloned into pmirGLO (E1330, Promega, Madison, WI, USA) at the polyclonal sites downstream of the luciferase gene (luc2) and named pIRAK1-WT (AGUUCUC). Site-directed mutagenesis was performed on the binding site between miR-146a-5p, and the target genes were predicted through bioinformatics analysis, followed by construction of the pIRAK1-MUT (UCAAGAG) vector. The steps were also applicable for TRAF6. miR-146a-5p mimic and NC were co-transfected with the luciferase reporter vector into human embryonic kidney HEK293T cells (iCell-h237, iCell Bioscience Inc., Shanghai, China). Using a luciferase assay kit (E1900, Promega, Madison, WI, USA), luciferase activity, as normalized to renilla luciferase, was detected by dual-luciferase reporter gene assay system (Dual-Luciferase^®^ Reporter Assay System, E1910, Promega, Madison, WI, USA).

### 2.12. RT-qPCR

Trizol (16096020, Invitrogen) was used for total cellular RNA extraction. For mRNA detection, a reverse transcription kit (11483188001, Roche, Basel, Switzerland) was used to obtain cDNA. For miRNA, a Polyad tailing kit (B532451, Sangon, Shanghai, China) was used to obtain the cDNA of miRNA containing PolyA tail. The PCR was carried out using the LightCycler 480 SYBR Green I Master. With GAPDH and U6 as the internal references of mRNA and miRNA, respectively, 2^−ΔΔCt^ method was adopted for quantitative analysis for gene expression. The primer sequences are shown in Appendix A.

### 2.13. Western Blot Analysis

Total protein of tissue or cells was extracted from a high-efficiency RIPA lysis buffer (Sigma Aldrich) containing 1% protease inhibitor and 1% phosphorylase inhibitor (Beyotime, Shanghai, China). Concentration of the extracted protein was determined using a BCA kit (Thermo Fisher Scientific, Waltham, MA, USA). The proteins were separated by polyacrylamide gel electrophoresis, then transferred to PVDF membranes (Millipore, Billerica, MA), and blocked with 5% BSA for 1 h at room temperature. The membranes were incubated with rabbit anti-mouse primary antibodies against CTNNB1 (1:500, ab68183, Abcam) and GAPDH (1:500, ab8245, Abcam) at 4 °C overnight. The next day, the membranes were further incubated with horseradish peroxidase-labeled goat anti-rabbit IgG (1:20,000, ab205718, Abcam) or goat anti-mouse IgG (1:20,000, ab197767, Abcam) diluent for 1.5 h at room temperature, followed by development using developing solution (NCI4106, Pierce, Rockford, IL, USA). Protein quantification analysis was performed by ImageJ 1.48 u software (2014, Bio-Rad Laboratories, Hercules, CA, USA) with an internal reference GAPDH.

### 2.14. Rat Model of Alveolar Bone Defects and In Vivo Experiment Protocols

In total, 48 male Wistar rats (6 weeks old, Vital River Laboratory Animal Technology Co., Ltd., Beijing, China) were raised under SPF environment, with laboratory humidity of 60–65%, temperature of 22–25 °C, and free access to food and water under 12 h light/dark cycles. The experiment was started after the rats were acclimatized for one week. After general anesthesia in rats, the alveolar bone was exposed by medical incision, and the model of alveolar bone defects was established in the second mandible molar (length: 3 mm; width: 1.5 mm; depth 1.5 mm) [25].

PKH26 was used to label Exos. Except for the rats without any treatment in the blank group, the alveolar bone defects of model rats were implanted with scaffolds of PF127 hydrogel, PF127 hydrogel@BMSC-NC-Exos (PF127 hydrogel loaded with BMSC-NC-Exos), or PF127 hydrogel@BMSC-CTNNB1-Exos (PF127 hydrogel loaded with BMSC-Exos containing CTNNB1). Rats were sacrificed with CO_2_ asphyxiation at 8 weeks after surgery, with the entire alveolar removed for micro-CT scanning. All specimens were fixed with 4% paraformaldehyde solution for 48 h.

### 2.15. Two-Photon Excited Fluorescence (TPEF) Imaging

To check the defect site for the presence of Exos, we randomly anesthetized three rats on day 3, day 14, day 28, and day 56 after the PF127 hydrogel@BMSC-Exos implantation. Next, we collected the alveolar bone tissue specimens from the implant site, and fluorescence was collected using the TPEF (Ni-E-A1RMP) superimposed scanning method with a 14 laser at wavelength of 1080 nm for PKH26 imaging. Finally, the signal intensities were analyzed using Image J software.

### 2.16. Immunohistochemistry

After antigen retrieval and normal goat serum blocking, the sections of rat alveolar bone tissues were incubated at 4 °C overnight with rabbit anti-mouse primary antibodies against RUNX2 (SAB1403638, 1:500, Sigma-Aldrich, St. Louis, MO, USA), OCN (AB10911, 1:500, Sigma-Aldrich), IRAK1 (SAB4504245, 1:200, Sigma-Aldrich, St. Louis, MO, USA), and TRAF6 (ab137452, 1:100, Abcam). The next day, the sections were incubated with goat anti-rabbit IgG (ab6721, 1:1000, Abcam) at 37 °C for 20 min and then with horseradish peroxidase-labeled streptomyces ovalbumin working solution (Imunbio, Beijing, China) at 37 °C for 20 min, followed by DAB (Whiga, Guangzhou, China) color development. Hematoxylin (Shanghai Bogoo Biological Technology Co., Ltd., Shanghai, China) was applied for counterstaining the sections. Images were finally observed and photographed under a microscope.

### 2.17. Micro-CT Scanning of Newly Formed Bone Tissue

Newly formed bone tissue at the rat alveolar bone defect was detected using a micro-CT scanner (SCANCO μCT50, Muttenz, Switzerland). Mimics software (Mimics 17.0, 2017, Materialise, Leuven, Belgium) was used to obtain 3D images and to calculate the bone mineral density (BMD) of the new bone-like tissue, and the relative trabecular bone volume [(bone volume (BV)/tissue volume (TV)] represented by the ratio between bone surface area and TV.

### 2.18. Statistical Analysis

Statistical analysis of the study data was performed using SPSS 21. 0 (IBM Corp. Armonk, NY, USA). Measurement data were expressed as mean ± standard deviation. Firstly, normality and homogeneity of variance were tested. Data between two groups obeying normal distribution and homogeneity of variance were compared with unpaired *t*-tests and those among multiple groups were compared by one-way ANOVA or repeated measures of ANOVA at different timepoints, followed by Tukey’s post hoc test. *p* < 0.05 indicated a statistically significant difference.

## 3. Results and Discussion

MSCs are widely used in bone tissue engineering due to their multipotential differentiation ability [26]. In recent years, increasing evidence has shown that transplanted MSCs exert their therapeutic effects through paracrine cytokines, rather than through direct cellular replacement, with Exos playing an important role [27]. The application potential of BMSC-Exos in bone regeneration was demonstrated by several studies, but the specific molecular mechanisms remain unclear [28,29]. Therefore, this study aimed to screen out BMSC-Exos-regulated key molecules in promoting bone regeneration and to further explore the downstream molecular mechanisms. The PF-127-containing scaffold is a promising candidate to accelerate bone tissue growth into the porous scaffold with good biocompatibility, which can provide a selection for bone regeneration in bone defects in the dental field [30]. Understanding the molecular mechanisms underlying the effect of PF127 hydrogel loaded with BMSC-Exos on bone regeneration provides novel strategies for alveolar bone defects.

### 3.1. CTNNB1 Might Be the Key Gene for BMSC-Exos to Promote the Differentiation of BMSCs into Osteoblasts

We retrieved proteomic data related to BMSC-Exos through the PRIDE partner database and obtained the PXD020948 project, which used ultracentrifugation to isolate Exos from MSCs and detect proteins using a marker-free method. Next, we extracted 771 BMSC-Exos-contained proteins in this project. In addition, the osteogenic differentiation of BMSCs-related expression profile data was retrieved through the GEO database, and the GSE9451 dataset was downloaded; 1151 genes significantly upregulated in the osteogenic differentiation of BMSCs were then screened using log*FC* > 1 and *p* < 0.05 as the thresholds (Figure 2A,B). The results from the PXD020948 and GSE9451 datasets were further intersected to obtain 54 intersecting genes (Figure 2C). The above intersecting genes were imported into the STRING database for protein interaction analysis and visualization and were sorted according to the number of intergenic connections (Degree value), which found that CTNNB1 ranked first (Figure 2D,E). In addition, evidence existed reporting that upregulated CTNNB1 expression promotes the osteogenic differentiation of MSCs [31,32]. Therefore, we speculated that CTNNB1 might be a key gene for BMSC-Exos in promoting the differentiation of BMSCs to osteoblasts.

### 3.2. Characterization of BMSC−CTNNB1−Exos

To explore the role of CTNNB1 in promoting bone regeneration by BMSC-Exos, we first constructed BMSCs with stable overexpression of CTNNB1. The transfection efficiency of CTNNB1 was verified by RT-qPCR and Western blot analysis (Figure 3A,B).

The BMSC-NC-Exos and BMSC-CTNNB1-Exos extracted by overspeed centrifugation showed typical cup morphology (Figure 3C). NTA found that the sizes of MSC-NC-Exos and BMSC-CTNNB1-Exos were mainly around 105 nm (Figure 3D), consistent with the diameters of Exos that were previously reported [33,34]. In addition, Western blot analysis found positive expression of TSG101 and CD9 on the surface of MSC-NC-Exos and BMSC-CTNNB1-Exos, while the calnexin was hardly expressed, indicating successful isolation of Exos (Figure 3E).

The Western blot analysis results showed that the expression of CTNNB1 was significantly increased after co-culture with BMSC-CTNNB1-Exos (Figure 3F), demonstrating that BMSC-CTNNB1-Exos with high CTNNB1 expression were successfully obtained.

### 3.3. PF127 Hydrogel Loaded with BMSC-Exos Efficiently Delivered CTNNB1 to BMSCs

Despite the great potential of MSC-derived Exos for therapeutic administration, there remains the problem of its low delivery efficiency and limited use in clinical research. In recent years, several strategies to promote the release of Exos have been developed, among which the loading of Exos by hydrogel is a safe and effective new treatment method [35]. PF-127 has a unique thermal sensitivity, in that it exists as a liquid at low temperatures and as a semi-solid gel at high temperatures [36]. For example, PF-127, which could gelate within a short response time and control drug release, was suggested as a novel approach for treating chronic periodontitis [37]. Due to this reversible thermal response behavior, PF-127 is able to adapt to a complex trauma environment, enabling the bioactive agent to adhere to the target and exert its biological effects [33]. Therefore, we attempted to load the BMSC-Exos with PF127 to form a complex, namely, PF127 hydrogel@BMSC-Exos.

The temperature sensitivity results showed that the initial gelation temperature of PF127 hydrogel@BMSC-Exos decreased as the PF127 concentration increased. Moreover, 20% PF127 hydrogel@BMSC-Exos composite was initially gelated at 18.1 °C, and 32% PF127 hydrogel@BMSC-Exos composite was initially gelated at 12.0 °C. Inversely proportional to the concentration, the gelation time of the 8% composite was only slightly different from that of the 32% composite at 37 °C. Therefore, we chose the 28% PF127 hydrogel@BMSC-Exos composite, which was liquid at 4 °C, was in a semi-solid colloid state at 37 °C, and could be gelated at 37 °C in about 43 s.

An increasing number of studies have reported the role of BMSC-derived Exos in bone repair or regeneration [28,29]. Exos from human gingiva-derived MSCs treated with TNF-α were reported to result in the inhibition of periodontal bone loss [38]. Exos secreted by human exfoliated deciduous teeth-derived stem cells could promote the repair of alveolar bone defects partially by regulating osteogenesis [39]. BMSC-derived small EVs hydrogel facilitated periodontal regeneration, as evidenced by the resultant less alveolar bone loss [14]. Intriguingly, a bioglass scaffold loaded with MSCs could facilitate alveolar bone repair in rhesus monkeys [40]. Simvastatin-loaded PF127 hydrogel had therapeutic efficacy on periodontal bone preservation in rats with ligature-induced periodontitis [41].

Subsequently, as observed by scanning electron microscopy (SEM), the loading of CTNNB1 did not significantly alter the morphology of PF127 hydrogel@BMSC-Exos, suggesting a relatively stable morphology of PF127 hydrogel@BMSC-Exos (Figure 4A). In addition, for further detection of the long-term stability of PF127 hydrogel@BMSC-CTNNB1-Exos and PF127 hydrogel@BMSC-Exos, we observed the morphological changes in the two after storage at −80 °C for 15 days and 30 days by SEM (Appendix A). The results showed that both PF127 hydrogel@BMSC-CTNNB1-Exos and PF127 hydrogel@BMSC-Exos maintained a good morphology at 30 days, which did not differ significantly from the morphology on the first day. The above results confirmed the good stability of the prepared PF127 hydrogel@BMSC-CTNNB1-Exos and PF127 hydrogel@BMSC-Exos.

To observe the uptake of PF127 hydrogel@BMSC-Exos by BMSCs, we used PKH67 (green fluorescence)-labeled Exos, with PF127 hydrogel as an NC and BMSC-Exos as a positive control. After 12 h of incubation with BMSCs, green fluorescence was shown in the presence of BMSCs but was not detected in response to PF127 hydrogel in the absence of BMSCs, suggesting the existence of Exos signal in the cytoplasm and the enrichment of it in the nucleus. Moreover, PF127 hydrogel brought about more enriched PKH67 green fluorescence in the presence of BMSC-NC-Exos or BMSC-CTNNB1-Exos (Figure 4B). These data suggest that the loading of Exos by PF127 hydrogel increased the uptake of Exos by BMSCs at the same concentration of Exos. The Western blot analysis results revealed that the expression of CTNNB1 in response to PF127 hydrogel@BMSC-NC-Exos or PF127 hydrogel@BMSC-CTNNB1-Exos was significantly increased when compared with that after treatment with PF127 hydrogel. The CTNNB1 expression was also notably elevated by PF127 hydrogel@BMSC-NC-Exos. The upregulation of CTNNB1 was particularly significant in the presence of PF127 hydrogel@BMSC-CTNNB1-Exos (Figure 4C,D). Collectively, PF127 hydrogel loaded with BMSC-Exos could better deliver the CTNNB1 to the BMSCs.

### 3.4. PF127 Hydrogel Loaded with BMSC-CTNNB1-Exos Promoted the Osteogenic Differentiation of BMSCs

Next, we further evaluated the effect of PF127 hydrogel-loaded BMSC-CTNNB1-Exos on the osteogenic differentiation of BMSCs. ALP staining (Figure 5A), Alizarin red staining (Figure 5B), and RT-qPCR (Figure 5C,D) illustrated that in both the PM and the OM, BMSC-NC-Exos or BMSC-CTNNB1-Exos enhanced ALP staining intensity and activity and extracellular matrix mineralization based on PF127 hydrogel treatment, accompanied by upregulated RUNX2 and OCN expression. The application of PF127 hydrogel augmented the ALP staining intensity and activity and extracellular matrix mineralization, while upregulating RUNX2 and OCN expression in BMSCs co-cultured with Exos, whether carrying CTNNB1 or not. This might be attributed to the elevated release of Exos by PF127 hydrogel. Compared to PF127 hydrogel@BMSC-NC-Exos, PF127 hydrogel@BMSC-CTNNB1-Exos resulted in increases in ALP staining intensity and activity and extracellular matrix mineralization of BMSCs as well as increases in RUNX2 and OCN expression. The changes in the above indicators were found to be more significant in the OM, indicating that overexpression of CTNNB1 significantly increased the osteogenic differentiation of BMSCs (Figure 5A–D). Therefore, PF127 hydrogel loaded with BMSC-CTNNB1-Exos could induce the osteogenic differentiation of BMSCs.

The increased expression of CTNNB1 by diosgenin could augment bone formation to exert anti-osteoporotic function [16]. CTNNB1 signaling regulated by iPTH enhanced osteoblastic differentiation to diminish alveolar bone loss during orthodontic tooth movement in a rat model of periodontitis [42]. The activated Wnt/CTNNB1 pathway by human amnion-derived MSCs promoted the osteogenic differentiation of BMSCs [43]. These studies concur with our finding regarding the role of CTNNB1 in the osteogenic differentiation of BMSCs and repair of alveolar bone defects; however, they failed to explore the related downstream regulatory mechanism of CTNNB1.

### 3.5. CTNNB1 Activated the Transcription of miR-146a-5p and Further Targeted Inhibited IRAK1 and TRAF6 during Osteogenic Differentiation of BMSCs

CTNNB1, which functions as a transcription factor, was suggested to activate the transcription of miR-146a [17]. It was also reported that miR-146a-5p is involved in the osteogenic differentiation of MSCs [18]. We designed two CTNNB1 shRNA sequences, and sh-CTNNB1-2 with optimal knockdown efficiency as determined by RT-qPCR was used for the subsequent ChIP assay (Appendix A). The ChIP results showed that the enrichment of CTNNB1 at the miR-146a-5p promoter region was significantly reduced after CTNNB1 knockdown (Figure 6A). The RT-qPCR results displayed that the expression of CTNNB1 and miR-146a-5p gradually increased on day 0, 7, and 14 (Figure 6B). Moreover, CTNNB1 and miR-146a-5p expression was upregulated by treatment with oe-CTNNB1 (Figure 6H). The above results indicated that CTNNB1 might act as a transcription factor to activate the transcription of miR-146a-5p and then promote its expression during the osteogenic differentiation of BMSCs.

We further explored the downstream pathway of miR-146a-5p and predicted the downstream target genes of miR-146a-5p using the miRDB (Target Score ≥ 90), TargetScan (Total context++ score ≤ −0.30), and DIANA TOOLS (miTG score ≥ 0.95) databases. The results of the three databases were intersected, obtaining 10 genes (TRAF6, IRAK1, ZBTB2, SLC10A3, CD80, EIF4G2, SIAH2, SEC23IP, BCORL1, and WWC2) (Figure 6C). Among them, only IRAK1 and TRAF6 had significantly differential expression during the osteogenic differentiation of BMSCs, both with a trend of gradual decrease (Figure 6D).

Evidence was previously demonstrated suggesting that BMSC-Exos-derived miRNA delays the development of the intervertebral disc degeneration by targeting TRAF6 [44]. Another prior study indicated that miR-146a-5p overexpression inhibited the IRAK1/TRAF6/NF-κB pathway in acute pancreatitis [45]. miR-146a-5p could simultaneously target IRAK1 and TRAF6 to inhibit their expression [19], but its role in the osteogenic differentiation of BMSCs has been rarely explored. According to the specific binding sites predicted by the TargetScan database (Figure 6E), the targeted binding relationships between miR-146a-5p and IRAK1 and between miR-146a-5p and TRAF6 were further verified by a dual-luciferase reporter gene assay. The results revealed that miR-146a-5p had a relatively good targeting relationship with IRAK1 and TRAF6 (Figure 6F,G, respectively). Thus, we propose that miR-146a-5p may specifically inhibit both IRAK1 and TRAF6 expression during the osteogenic differentiation of BMSCs.

To elucidate the regulatory relationship of the CTNNB1-miR-146a-5p/IRAK1/TRAF6 signaling axis during the osteogenic differentiation of BMSCs, we interfered CTNNB1 and miR-146a-5p expression in BMSCs and examined the downstream factor expression changes by RT-qPCR. The results showed that the expression of IRAK1 and TRAF6 was significantly downregulated in the presence of CTNNB1 overexpression; additional treatment with miR-146a-5p inhibitor failed to alter CTNNB1 expression but diminished the expression of miR-146a-5p while upregulating that of IRAK1 and TRAF6 (Figure 6H), indicating that the knockdown of miR-146a-5p expression restored the inhibition of IRAK1 and TRAF6 expression by CTNNB1 overexpression. Collectively, both CTNNB1 and miR-146a-5p were upregulated, while IRAK1 and TRAF6 were downregulated, during the osteogenic differentiation of BMSCs. CTNNB1 might further target and inhibit the expression of IRAK1 and TRAF6 by activating the transcription of miR-146a-5p.

### 3.6. CTNNB1 Induced the Osteogenic Differentiation of BMSCs by Regulating the miR-146a-5p/IRAK1/TRAF6 Axis

We then explored whether CTNNB1 promoted the osteogenic differentiation of BMSCs through the subsequent targeted inhibition of IRAK1 and TRAF6 expression by activating the transcription of miR-146a-5p. Two different shRNA sequences, targeting IRAK1 and TRAF6, respectively, were designed, with sh-IRAK1-1 and sh-TRAF6-2 showing optimal knockdown results by RT-qPCR (Appendix A) and, thus, being selected for further analyses.

The ALP staining (Figure 7A), Alizarin red staining (Figure 7B), and RT-qPCR (Figure 7C,D) results showed that, in both PM and OM, overexpression of CTNNB1 led to enhanced ALP activity and extracellular matrix mineralization, accompanied by upregulated RUNX2 and OCN, indicating that overexpression of CTNNB1 induced the osteogenic differentiation of BMSCs. In contrast to CTNNB1 overexpression alone, CTNNB1 overexpression combined with miR-146a-5p inhibition contributed to notably reduced ALP activity and extracellular matrix mineralization and downregulated RUNX2 and OCN expression, suggesting that knockdown of miR-146a-5p could restore the induction of the osteogenic differentiation of BMSCs by CTNNB1 overexpression. Moreover, additional knockdown of IRAK1/TRAF6 restored the inhibitory effect of knockdown of miR-146a-5p on the osteogenic differentiation of BMSCs, as evidenced by the increased ALP activity and extracellular matrix mineralization and upregulated RUNX2 and OCN expression vs. those in response to CTNNB1 overexpression combined with miR-146a-5p inhibition (Figure 7A–D). The above results suggested that CTNNB1 might activate the transcription of miR-146a-5p and then inhibit IRAK1 and TRAF6 expression, ultimately promoting the osteogenic differentiation of BMSCs.

### 3.7. PF127 Hydrogel Loaded with BMSC-CTNNB1-Exos Induced Alveolar Bone Regeneration in Rats

We have already confirmed that PF127 hydrogel loaded with BMSC-CTNNB1-Exos had a good temperature sensitivity and release efficiency and could significantly promote the osteogenic differentiation of BMSCs. Therefore, we further explored the role of PF127 hydrogel loaded with BMSC-CTNNB1-Exos in the repair of alveolar bone defects in rats.

We successfully constructed a rat model of alveolar bone defects. First, we implanted PKH26-labeled PF127 hydrogel@BMSC-Exos into the defect site. The in vivo TPEF imaging results showed that the Exos were distributed in the defect site 3 d after PF127 hydrogel implantation and lasted for at least 14 d (Figure 8A), demonstrating a good implantation effect. The micro-CT scanning images and the quantitative results showed no significant difference in the new bone formation, BV/TV ratio, or BMD after treatment with PF127 hydrogel. In contrast to PF127 hydrogel, PF127 hydrogel@BMSC-NC-Exos led to increased new bone formation, accompanied by an elevated BV/TV ratio and BMD, which could be further elevated in the presence of CTNNB1 overexpression (Figure 8B). In addition, the immunohistochemical staining results showed no significant difference in the positive expression of RUNX2, OCN, IRAK1, and TRAF6 after treatment with PF127 hydrogel. In contrast, treatment with PF127 hydrogel@BMSC-NC-Exos resulted in a higher staining particle range and intensity for RUNX2 and OCN around the nucleus and within the cytoplasm of the osteoblasts, while IRAK1 and TRAF6 displayed lower intensity; these effects could be furthered in response to additional CTNNB1 overexpression (Figure 8C). Therefore, PF127 hydrogel loaded with BMSC-CTNNB1-Exos could induce alveolar bone regeneration and promote the repair of alveolar bone defects in rats, which might be related to the downregulation of the downstream genes IRAK1 and TRAF6.

CTNNB1, serving as a transcription factor, was previously revealed to activate miR-146a transcription [17]. Furthermore, overexpression of miR-146-5p activated Wnt/β-catenin in lung cancer cells [46]. As previously reported, the osteogenic Exos secreted by MSCs could deliver upregulated hsa-miR-146a-5p to induce osteogenic differentiation, presenting the potential of being applied in a bone regeneration strategy [47]. In addition, overexpression of miR-146a-5p could accelerate the osteogenic differentiation of human placenta-derived MSCs, which may improve the osteogenic efficacy in bone defect repair using scaffold materials [48]. Exosomal miR-146a-5p from human urine-derived stem cells could target and downregulate IRAK1 to alleviate renal ischemia/reperfusion injury in rats [49]. Moreover, miR-146a-5p delivered by human umbilical cord MSCs repressed TRAF6 signaling to augment protection against rat diabetic nephropathy via M2 macrophage polarization [50]. Of note, previous research also unfolded the involvement of the IRAK1-TRAF6 in bone-related diseases. For instance, the activated TLR2/4-IRAK1/TRAF6 axis due to glycemic fluctuation aggravated bone loss [20]. Additionally, TRAF6 inhibition by treatment with astragalus polysaccharide aided in protecting the alveolar bone, which is involved with the decline in local osteoclasts [51]. Therefore, it is suggested in our study that CTNNB1 activated the transcription of miR-146a-5p and then downregulated the expression of IRAK1 and TRAF6 during the osteogenic differentiation of BMSCs.

## 4. Conclusions

To conclude, this study demonstrated that CTNNB1 overexpression inhibits IRAK1 and TRAF6 expression by activating miR-146a-5p transcription, which makes it possible for PF127 hydrogel loaded with BMSC-CTNNB1-Exos to promote the osteogenic differentiation of BMSCs, thereby facilitating the repair of alveolar bone defects in rats (Figure 1). This finding may provide a promising technical means for the repair of alveolar bone defects and further the understanding of the molecular mechanism of alveolar bone regeneration. Nevertheless, the clinical feasibility still needs further validation.

## Figures and Tables

**Figure 1 nanomaterials-13-01083-f001:**
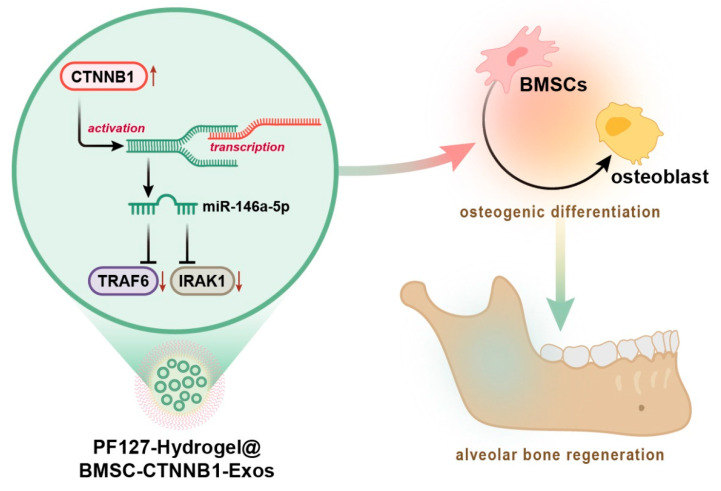
Molecular mechanism graph delineating the role of directed differentiation of BMSCs induced by BMSC-CTNNB1-Exos-loaded PF127 hydrogel in alveolar bone regeneration. CTNNB1 targets and inhibits IRAK1 and TRAF6 expression by activating the transcription of miR-146a-5p. PF127 hydrogel-loaded BMSC-CTNNB1-Exos promote osteogenic differentiation of BMSCs and alveolar bone regeneration, thereby facilitating repair of alveolar bone defects in rats.

**Figure 2 nanomaterials-13-01083-f002:**
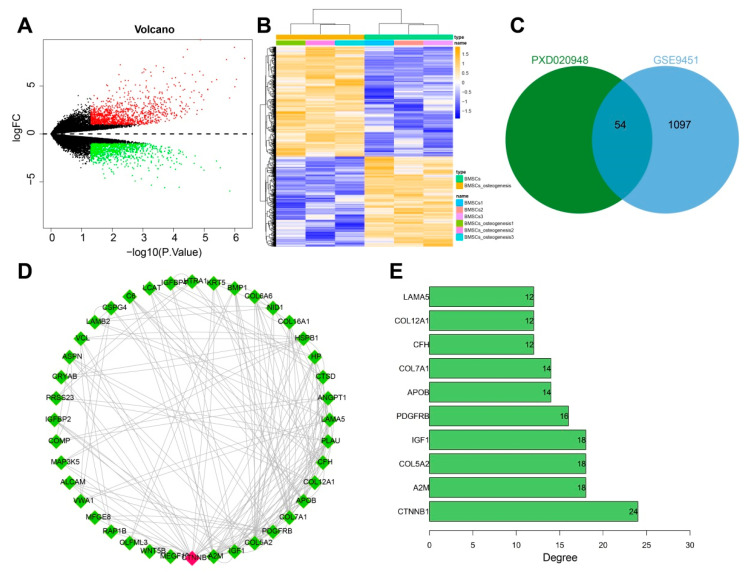
Screening of key genes for BMSC–Exos to promote BMSC osteogenic differentiation. (**A**) The volcano plot for differential analysis of GSE9451 dataset. Red dots indicate significantly upregulated genes, green dots indicate significantly downregulated genes, and black dots indicate genes not differentially expressed (BMSCs group: *n* = 3; BMSCs osteogenesis group: *n* = 3). (**B**) Heatmap for differential analysis of GSE9451 dataset (BMSCs group: *n* = 3; BMSCs osteogenesis group: *n* = 3). (**C**) Venn diagram for the intersection of PXD020948 and GSE9451 datasets. (**D**) Protein interaction diagram of the 54 candidate genes, in which the counterclockwise degree values were sequentially decreased from CTNNB1. (**E**) The top 10 genes displayed according to the degree values.

**Figure 3 nanomaterials-13-01083-f003:**
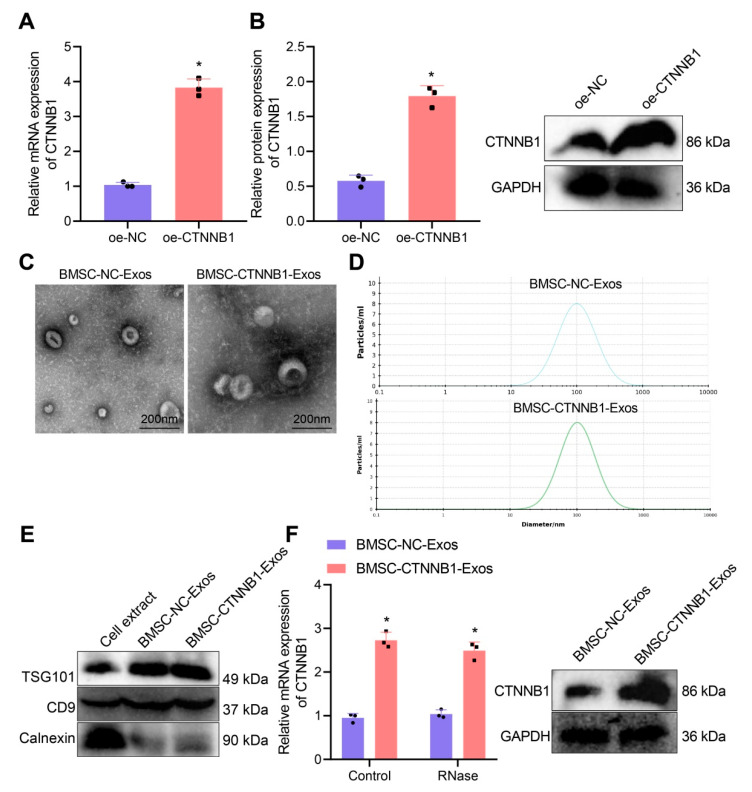
Isolation and characterization of the BMSC-NC-Exos and BMSC-CTNNB1-Exos. (**A**) The expression of CTNNB1 in the BMSC-NC-Exos and BMSC-CTNNB1-Exos detected by RT-qPCR. (**B**) Western blot analysis for the expression of CTNNB1 in the BMSC-NC-Exos and BMSC-CTNNB1-Exos. (**C**) The morphology of the BMSC-NC-Exos and BMSC-CTNNB1-Exos. (**D**) NTA showing the particle size of the BMSC-NC-Exos and BMSC-CTNNB1-Exos. (**E**) Western blot analysis of the protein expression of TSG101, CD9, and Calnexin in cell extracts as well as BMSC-NC-Exos and BMSC-CTNNB1-Exos. (**F**) Western blot analysis for detection of the protein expression of CTNNB1 in the BMSC-NC-Exos and BMSC-CTNNB1-Exos. * *p* < 0.05 vs. the oe-NC group, the cell extract group or the BMSC-NC-Exos group. Cell experiments were independently repeated three times.

**Figure 4 nanomaterials-13-01083-f004:**
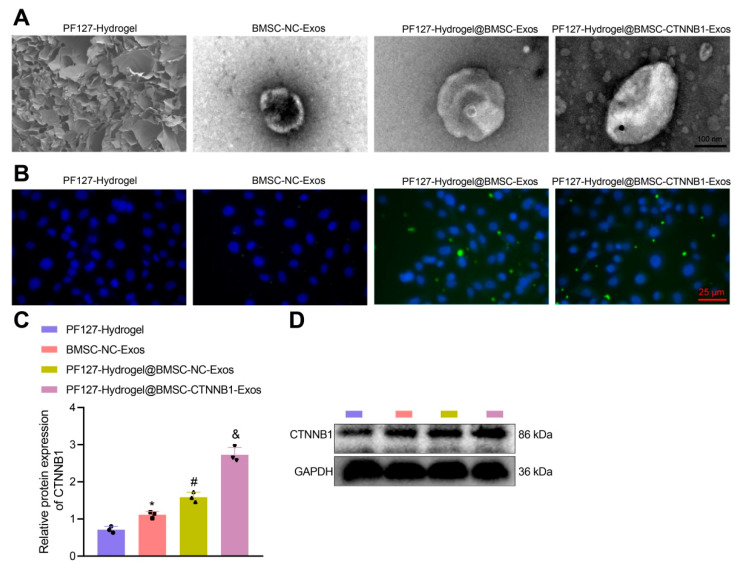
Assessment of uptake of PF127 hydrogel@BMSC-Exos by BMSCs. (**A**) The morphology of PF127 hydrogel, BMSC-NC-Exos, PF127 hydrogel@BMSC-Exos, and PF127 hydrogel@BMSC-CTNNB1-Exos, as observed by SEM. (**B**) The PKH67 labeling method to detect the uptake of Exos by BMSCs in response to PF127 hydrogel@BMSC-CTNNB1-Exos. (**C**,**D**) The protein expression of CTNNB1 in BMSCs in response to PF127 hydrogel@BMSC-CTNNB1-Exos, as determined by Western blot analysis. * *p* < 0.05 vs. the PF127 hydrogel group. # *p* < 0.05 vs. the BMSC-NC-Exos group. & *p* < 0.05 vs. the PF127 hydrogel@BMSC-NC-Exos group. Cell experiments were independently repeated three times.

**Figure 5 nanomaterials-13-01083-f005:**
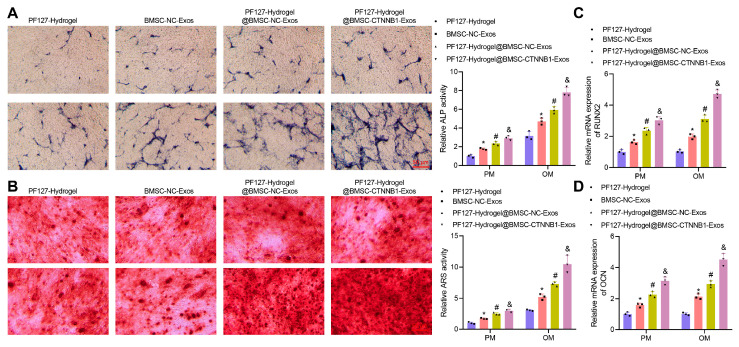
Effects of the PF127 hydrogel-loaded BMSC-CTNNB1-Exos on the osteogenic differentiation of the BMSCs. (**A**) ALP staining and quantitative results of BMSCs cultured with PM and OM in response to PF127 hydrogel@BMSC-CTNNB1-Exos. (**B**) Alizarin red staining and quantitative results of the BMSCs cultured with PM and OM in response to PF127 hydrogel@BMSC-CTNNB1-Exos. (**C**) RT-qPCR detection of RUNX2 expression in BMSCs cultured with PM and OM in response to PF127 hydrogel@BMSC-CTNNB1-Exos. (**D**) OCN expression in BMSCs cultured with PM and OM in response to PF127 hydrogel@BMSC-CTNNB1-Exos determined by RT-qPCR. * *p* < 0.05 vs. the PF127 hydrogel group. # *p* < 0.05 vs. the BMSC-NC-Exos group. & *p* < 0.05 vs. the PF127 hydrogel@BMSC-NC-Exos. Cell experiments were independently repeated three times.

**Figure 6 nanomaterials-13-01083-f006:**
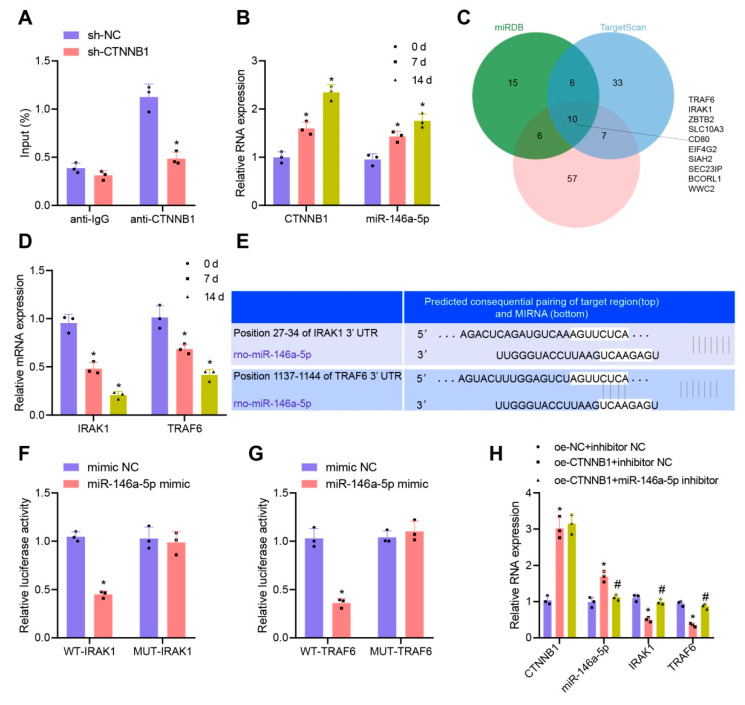
CTNNB1 activates the transcription of miR-146a-5p and further inhibits IRAK1 and TRAF6 during osteogenic differentiation of BMSCs. (**A**) ChIP for detection of the enrichment of CTNNB1 at the promoter region of the miR-146a-5p. (**B**) The CTNNB1 and miR-146a-5p expression after 0, 7, and 14 d of osteogenic differentiation of BMSCs determined by RT-qPCR. (**C**) Venn diagram for the intersection of miRDB, TargetScan, and DIANA TOOLS databases. (**D**) IRAK1 and TRAF6 expression after 0, 7 and 14 d determined by RT-qPCR. (**E**) The targeted binding sites between miR-146a-5p and IRAK1 and between miR-146a-5p and TRAF6 predicted through the TargetScan database. (**F**) Dual-luciferase reporter gene assay to verify the targeted binding relationship between miR-146a-5p and IRAK1. (**G**) Dual-luciferase reporter gene assay to verify the targeted binding relationship between miR-146a-5p and TRAF6. (**H**) Expression of CTNNB1, miR-146a-5p, IRAK1, and TRAF6 in BMSCs upon CTNNB1 overexpression alone or combined with miR-146a-5p inhibition detected by RT-qPCR. * *p* < 0.05 vs. the sh-NC, 0 d, mimic NC, or oe-NC + inhibitor NC group. # *p* < 0.05 vs. the oe-CTNNB1 + inhibitor NC group. Cell experiments were independently repeated three times.

**Figure 7 nanomaterials-13-01083-f007:**
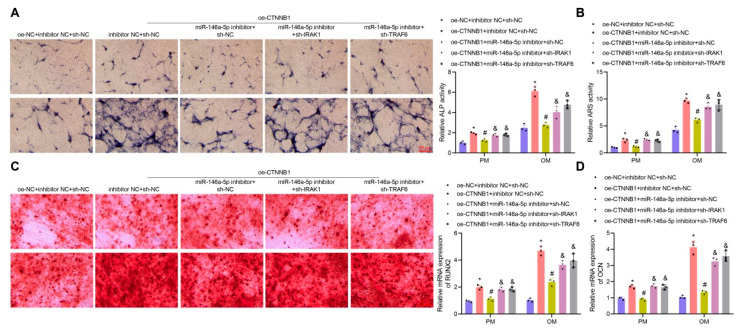
CTNNB1 induces the osteogenic differentiation of BMSCs by regulating the miR-146a-5p/IRAK1/TRAF6 axis. BMSCs were cultured with PM and OM and treated with oe-CTNNB1, miR-146a-5p inhibitor, sh-IRAK1, or sh-TRAF6. (**A**) ALP staining for BMSCs and the quantitative results. (**B**) Alizarin red staining for BMSCs and quantitative results. (**C**) RT-qPCR detection of RUNX2 expression in BMSCs. (**D**) OCN expression in BMSCs determined by RT-qPCR. * *p* < 0.05 vs. the oe-NC + inhibitor NC + sh-NC group. # *p* < 0.05 vs. the oe-CTNNB1 + inhibitor NC + sh-NC group. & *p* < 0.05 vs. the oe-CTNNB1 + miR-146a-5p inhibitor + sh-NC group. Cell experiments were independently repeated three times.

**Figure 8 nanomaterials-13-01083-f008:**
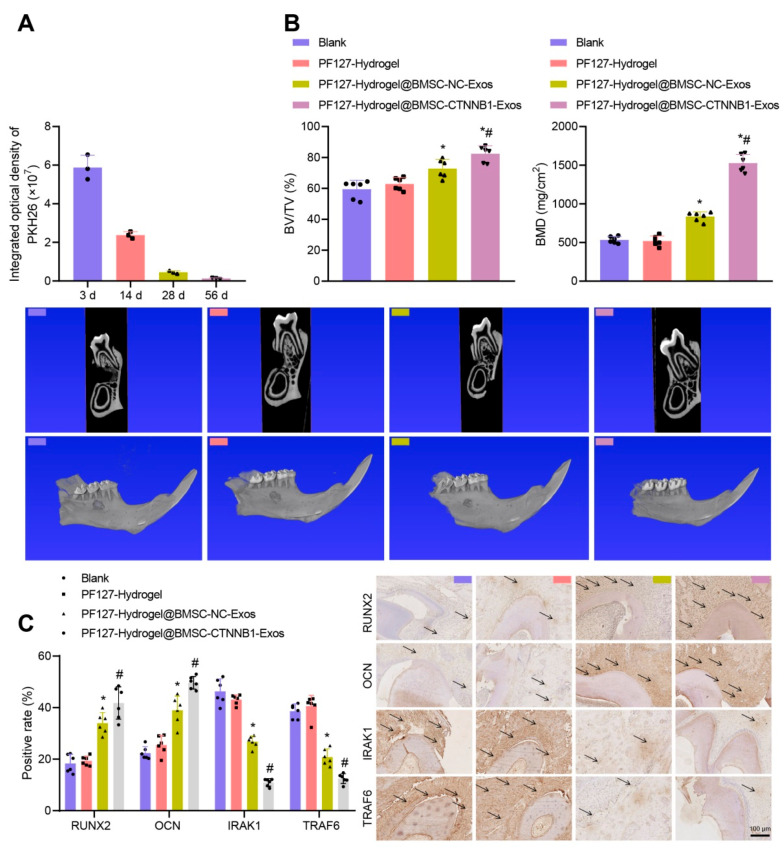
PF127 hydrogel loaded with BMSC-CTNNB1-Exos induces the alveolar bone regeneration in rats. (**A**) In vivo TPEF recording of the distribution of PKH26-labeled Exos implanted at different time points and quantitative analysis. *n* = 3. (**B**) The micro-CT scanning images focusing on the second mandibular molar (red frame) and the quantitative results of alveolar bone defects in rats treated with PF127 hydrogel@BMSC-CTNNB1-Exos. *n* = 6. (**C**) The positive expression of RUNX2, OCN, IRAK1, and TRAF6 in new bones detected by immunohistochemical staining. The black arrows indicate dark brown particles, which represent protein positive staining. *n* = 6. * *p* < 0.05 vs. the PF127 hydrogel group. # *p* < 0.05 vs. the PF127 hydrogel@BMSC-NC-Exos group.

## Data Availability

The data that support the findings of this study are available on request from the corresponding author. The current study was approved by the Animal Ethics Committee of our hospital. Extensive efforts were made to minimize both the number of animals and their respective suffering in the experiments.

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
