# Peer review of "PF127 Hydrogel-Based Delivery of Exosomal CTNNB1 from Mesenchymal Stem Cells Induces Osteogenic Differentiation during the Repair of Alveolar Bone Defects"

_nanomaterials, 2023, doi:10.3390/nano13061083_

Round 1

Reviewer 1 Report

The publication is a generally well written report covering an absolutely interesting topic. The research question is clearly framed and the literature is relevant. The authors wrote “Autologous bone transplantation has long been considered the “gold standard” for severe alveolar bone defects…”, but generally the critical size defect model is the gold-standard technique if we would like to evaluate the effect of different systems, scaffolds. The rat model of alveolar bone defect’s width is only 1.5 mm and the authors should support this protocol and have to evaluate the difference between the critical size defect model.

Author Response

Response to Reviewer 1

Reviewer 1

The publication is a generally well written report covering an absolutely interesting topic. The research question is clearly framed and the literature is relevant. The authors wrote “Autologous bone transplantation has long been considered the “gold standard” for severe alveolar bone defects…”, but generally the critical size defect model is the gold-standard technique if we would like to evaluate the effect of different systems, scaffolds. The rat model of alveolar bone defect’s width is only 1.5 mm and the authors should support this protocol and have to evaluate the difference between the critical size defect model.

Response: Thanks for your suggestion. We have revised the inappropriate description. Literature supporting the establishment of the rat alveolar bone defect model comes from a study by King et al. in J Dent Res (King GN, King N, Cruchley AT, Wozney JM, Hughes FJ. Recombinant human bone morphogenetic protein-2 promotes wound healing in rat periodontal fenestration defects. J Dent Res. 1997 Aug;76(8):1460-70. doi: 10.1177/00220345970760080801. PMID: 9240382.), in which an alveolar bone defect model is created centered on the second mandibular molar in the right mandibular buccal alveolar bone with a width of 1.5 mm and a depth of 1.5 mm. Because the drill bits we used in establishing the animal model were all 1.5 mm in width, the degree of defects remained consistent between the alveolar bone defect model animals, with no significant difference.

Reviewer 2 Report

The manuscript “PF127 hydrogel-based delivery of exosomal CTNNB1 from mesenchymal stem cells induces osteogenic differentiation during the repair of alveolar bone defects” deals with the production of Pluronic F127 (PF127) hydrogels for bone regeneration. Several analyses and characterizations of the samples were performed, obtaining intriguing results. However, some revisions are required, as follows:

- Abstract. Add quantitative results to this section.

- Introduction. The state of the art on the use of biomaterials for bone regeneration should be enlarged, focusing also on the preparation techniques. For this purpose, see for instance these works: Baldino et al., Polymer Engineering and Science, 2018, 58(9), pp. 1494–1499; Kang et al., Nanomaterials, 2021, 11(12), 3215; etc..

- R&D. SEM analysis could be added in order to observe the internal morphology of the samples. Moreover, a systematic comparison and discussion of the results with the previous literature could help the reader in understanding the relevance of the present work.

- Improve English.

Author Response

Response to Reviewer 2

Reviewer 2

The manuscript “PF127 hydrogel-based delivery of exosomal CTNNB1 from mesenchymal stem cells induces osteogenic differentiation during the repair of alveolar bone defects” deals with the production of Pluronic F127 (PF127) hydrogels for bone regeneration. Several analyses and characterizations of the samples were performed, obtaining intriguing results. However, some revisions are required, as follows:

- Abstract. Add quantitative results to this section.

Response: Thanks for your suggestion. We have added quantitative results to the Abstract section.

- Introduction. The state of the art on the use of biomaterials for bone regeneration should be enlarged, focusing also on the preparation techniques. For this purpose, see for instance these works: Baldino et al., Polymer Engineering and Science, 2018, 58(9), pp. 1494–1499; Kang et al., Nanomaterials, 2021, 11(12), 3215; etc..

Response: Thanks for your suggestion. We have cited the references recommended by you in Introduction: Baldino et al. developed the supercritical gel drying of Ch/G mixtures to produce aerogels with improved structural organization and properties in regard to the starting single biopolymers [1,7]. In addition, Kang et al. illustrated that the PTMC/PLA/HA and PTMC/HA scaffold can be prepared utilizing PTMC/PLA/HA and PTMC/HA composite materials via the biological 3D printing method and developed as potential biomaterials for bone repatriation and tissue engineering [8].

- R&D. SEM analysis could be added in order to observe the internal morphology of the samples. Moreover, a systematic comparison and discussion of the results with the previous literature could help the reader in understanding the relevance of the present work.

Response: Thanks for your suggestion. We have added SEM analysis to observe the internal morphology of the samples: Subsequently, as observed by scanning electron microscopy, loading of CTNNB1 did not significantly alter the morphology of PF127 hydrogel@BMSC-Exos, suggesting a relatively stable morphology of PF127 hydrogel@BMSC-Exos (Figure 4A). Besides, the discussion has been revised accordingly.

- Improve English.

Response: Thanks for your comment. We have invited a native English speaker to re-edit the manuscript. We hope that the improved language can meet your requirement.

Reviewer 3 Report

The presented paper is devoted to preparation of composite PF127-based hydrogel with entrapped exosomes, which were derived form bone marrow MSCs. Such exosomes bear CTNNB1 gene, which induces formation of microRNA-146a-5p, whic, in its turn, affect osteodifferentiation. The idea is actual and performed experiments will be interesting for scientists dealing with tissue engineering. Despite the general positive impression some issues should be met before publication.

1. There are many grammatical mistakes, which should be corrected. 

2. The lines are anscent in the MS file, which makes it harder to point out the parts of the text to be corrected.

3. The role of beta-catenin gene should be better explained. Ref. 19 should be better discussed within the context of current paper in order to clarify the logical chain to the purpose of presented study.

4. Please provide the Ethical Committee session protocol number.

5. Page 3. Part 2.6. doesn't contains essential information on the weight of PL127 taken for hydrogel preparation as well as on way of mixing? Was it vortex, thermoshaker, etc?

6. The stability of exosomes within hydrogel made of surface active agent (PF 127) was not assessed. Please provide the data, which proves that exosomes are stable within hydrogel.

7. In my opinion Fig. 8 will be much more useful if it will be Fig.1 and placed into introduction or very begining of R&D part, because it shows the overall strategy of this work.

Author Response

Response to Reviewer 3

Reviewer 3

The presented paper is devoted to preparation of composite PF127-based hydrogel with entrapped exosomes, which were derived form bone marrow MSCs. Such exosomes bear CTNNB1 gene, which induces formation of microRNA-146a-5p, whic, in its turn, affect osteodifferentiation. The idea is actual and performed experiments will be interesting for scientists dealing with tissue engineering. Despite the general positive impression some issues should be met before publication.

  1. There are many grammatical mistakes, which should be corrected.

Response: Thanks for your comment. The manuscript has been re-edited by a native English speaker. We hope that the improved language can meet your requirement.

  1. The lines are anscent in the MS file, which makes it harder to point out the parts of the text to be corrected.

Response: Sorry for the negligence. We have supplemented the lines in the MS file.

  1. The role of beta-catenin gene should be better explained. Ref. 19 should be better discussed within the context of current paper in order to clarify the logical chain to the purpose of presented study.

Response: Thanks for your suggestions. We have further explained the role of beta-catenin gene in Introduction: As a multitasking and evolutionary conserved molecule, β-catenin (Armadillo in Drosophila) in metazoans plays a pivotal role in multiple developmental and homeostatic processes and functions as an important nuclear effector of canonical Wnt signaling in the nucleus [15]. In addition, we have expanded the discussion on Ref. 19: Of note, Li et al. have demonstrated that more notable peri-implant bone loss and osteoclastogenesis were found in diabetic mice with glycemic fluctuation, and glycemic fluctuation could result in increases in expression of IRAK1 and TRAF6 in peri-implant gingival tissues, which suggested that activation of IRAK1-TRAF6 axis by glycemic fluctuation may contribute to aggravation of bone loss [20].

  1. Please provide the Ethical Committee session protocol number.

Response: Accordingly, we have provided the Ethical Committee session protocol number.

  1. Page 3. Part 2.6. doesn't contains essential information on the weight of PL127 taken for hydrogel preparation as well as on way of mixing? Was it vortex, thermoshaker, etc?

Response: Thanks for reminding us. We have supplemented the information on weight of PL127 taken for hydrogel preparation as well as on way of mixing in Page 3. Part 2.6: Pre-cooled PF-127 powder (P2443, Sigma-Aldrich) and 300 μg/mL of above BMSC-Exos suspension were mixed in a 1.5 mL centrifuge tube. The powder was fully dissolved by vortexing, followed by an ice bath and preservation in a 4°C refrigerator. Please check it.

  1. The stability of exosomes within hydrogel made of surface active agent (PF 127) was not assessed. Please provide the data, which proves that exosomes are stable within hydrogel.

Response: Based on your suggestion, we have provided the data, which confirmed that exosomes were stable within hydrogel: In addition, for further detection the long-term stability of PF127 hydrogel@BMSC-CTNNB1-Exos and PF127 hydrogel@BMSC-Exos, we observed the morphological changes of the two after storage at -80℃ for 15 days and 30 days through SEM (Supplementary Figure 1). The results showed that both PF127 hydrogel@BMSC-CTNNB1-Exos and PF127 hydrogel@BMSC-Exos remained good morphology at 30 day, which did not differ significantly from that on the first day. The above results confirmed the good stability of the prepared PF127 hydrogel@BMSC-CTNNB1-Exos and PF127 hydrogel@BMSC-Exos.

  1. In my opinion Fig. 8 will be much more useful if it will be Fig.1 and placed into introduction or very begining of R&D part, because it shows the overall strategy of this work.

Response: Based on your suggestion, we have placed the original Figure 8 to the end of Introduction, as current Figure 1.
